# Representative Ring Signature Algorithm Based on Smart Contract

**DOI:** 10.3390/s22186805

**Published:** 2022-09-08

**Authors:** Qiude Li, Wenlong Yi, Xiaomin Zhao, Hua Yin, Igor Gerasimov

**Affiliations:** 1School of Software, Jiangxi Agricultural University, Nanchang 330045, China; 2Faculty of Computer Science and Technology, Saint Petersburg Electrotechnical University “LETI”, 197022 Saint Petersburg, Russia; 3Key Laboratory of Poyang Lake Watershed Agricultural Resources and Ecology of Jiangxi Province, Jiangxi Agricultural University, Nanchang 330045, China

**Keywords:** ring signature, multiparty secure computation, consortium chain, smart contract

## Abstract

Traditional ring signature algorithms suffer from large signature data capacity and low speed of signature and verification during collective signing. In this work, we propose a representative ring signature algorithm based on smart contracts. By collecting the opinions of the signatory based on multiparty secure computation, the proposed technique protects the privacy of the signatory during the data interaction process in the consortium chain. Moreover, the proposed method uses smart contracts to organize the signature process and formulate a signature strategy of “one encryption per signature” to prevent signature forgery. It uses the Hyperledger Fabric framework as the signature test platform of the consortium chain to perform the experiments. We compare the results of the proposed method with the ECC ring signature scheme. The experimental results show that in the worst case, the signature volume of the proposed method decreases by more than two times, and the signature speed and verification speed increase by more than three times. Therefore, in the collective signature scenario of transaction verification in the consortium chain, the proposed method is verified to be innovative and practical.

## 1. Introduction

Blockchain is regarded as the foundation of the next-generation value Internet. After more than 10 years of development, the development of blockchain technology has gone through three stages, including programmable currency, programmable finance, and programmable society [1]. The essence of blockchain is based on a distributed ledger integrating cryptography, a consensus algorithm, and a P2P network [2]. It is noteworthy that blockchains are decentralized, immutable, and traceable [3,4,5]. From the data-structure point of view, blockchains are linked by blocks. A block consists of a block header and a block body. In the block header, the blockchain timestamps ensure that the transaction order is irreversible, the block hash maintains the immutability of the transaction record, and the Merkle root verifies the transaction consistency. In the block body, the serialized transactions form a Merkle tree and provide the data support for the Merkle root [6]. The first use of blockchain technology was digital currency. Since then, it has been applied in various other fields, such as supply chains, the Internet of things, intelligent manufacturing, and healthcare. Currently, blockchain plays a central role in industrial transformation and technological innovation. In the future, the “blockchain+” and “+blockchain” are expected to become a new business model [7,8,9]. Currently, the data for these applications are extracted using big data and aggregation analysis. These techniques may be at risk of privacy data leakage [10]. It is possible to store a wide variety of data, including financial information, identity, assets of an individual, etc., in a blockchain [11]. In the real world, as the confidentiality of data is a critical issue, the leakage of such data can have invaluable consequences [12]. Therefore, privacy security is a critical research topic in the development of blockchain technology [13,14,15].

Cryptography forms the basis of data security. Algorithms such as the Advanced Encryption Standard (AES) and elliptic curve cryptography (ECC) are widely used for securing data [16,17]. However, with the development of blockchain, traditional encryption methods are unable to meet the application requirements. In order to address this problem, there are various other methods proposed in literature, such as zero-knowledge proof [18], ring signature [19], and homomorphic encryption [20] methods. In terms of zero-knowledge proofs, in [21], the authors used a noninteractive zero-knowledge (NIZK) proof in cryptocurrency to ensure that the transaction was valid and the amount remained a secret. In [22], the authors used a zero-knowledge proof authentication scheme based on an elliptic curve in the medical system to avoid illegal theft of the patients’ data. In [23], the authors proposed a zero-knowledge scope proof in the Internet of vehicles, which realizes the geographic authentication of vehicles in the network and ensures the privacy security of vehicle owners. In terms of ring signatures, in [24], the authors used a ring signature in Monroe to confuse the transaction source, which protects the identity information of the transaction party. In [25], the authors introduced ring signatures in the voting system for implementing anonymous voting. In terms of homomorphic encryption, in [26], the authors adopted a homomorphic encryption scheme, which protects private data in smart homes. In [27], the authors proposed a distributed smart meter data-aggregation framework with blockchain and homomorphic encryption as the backbone, which significantly enhances the security and privacy protection of meter data aggregation. In [28], the authors introduced an edge computing and fully homomorphic Paillier cryptosystem, designed a distributed privacy protection architecture based on blockchain, and realized the security protection of cloud data. In these information protection schemes, the proof efficiency generated by zero-knowledge proof is low, and the homomorphic encryption has a high computational complexity. On the contrary, ring signatures provide sufficient privacy protection under few computing resources. Therefore, ring signature technology has important research and practical value in terms of information protection. Currently, ring signatures are commonly used to verify the legitimacy of transactions in a blockchain. As shown in Figure 1, when the transaction initiator submits a transaction, it needs to be digitally signed. In order to keep the identity information hidden, the initiator organizes the ring signature of the transaction. In the ring signature process, the private key of the initiator and the public keys of other members are collected. Then, the completed signature is uploaded in the blockchain. In the blockchain, the transactions are first broadcast across the network among peer nodes, and then the successfully verified transactions are written in a new block. During the whole process mentioned above, the ring signature and blockchain only show the combination between the hierarchical relationships, and do not achieve deep integration, so that the maximum advantages between the two cannot be exploited. Exploring more forms of application of ring signatures in blockchain and exploring the maximum potential of the combination of both will benefit the further development of blockchain.

## 2. Related Works

In 2001, Rivest et al. [19] first proposed the ring signature technology, which uses encryption to conceal a private key in a public key group. In this way, the verifier is unable to recognize the private key of the initiator. Inspired by this method, different researchers have improved and applied this method in blockchain. The public key cryptography still forms the core of ring signature. It is noteworthy that the construction of public key cryptography algorithms depends on mathematical notions. Therefore, the well-known improvement method of ring signatures is to replace the underlying mathematical problems. The main research can be divided into three categories, the including bilinear pairing problem [29], elliptic curve problem [30], and lattice problem [31]. In [32], the authors designed a ring signature scheme based on attribute revocation, which relies on the difficulty of solving the bilinear pair problem to ensure the security of the signature. In [33], the authors constructed a privacy data storage protocol based on the ring signature on the elliptic curve, which ensures the security of data and user identity privacy in the blockchain applications. In [34], the authors constructed a lattice based one-time linkable ring signature scheme, which verifies the unconditional anonymity and unforgeability of the transaction information under standard lattice hypothesis and provides information security under quantum attacks. Among the aforementioned three kinds of ring signature schemes, the bilinear pairing scheme has a fairly low security, and the current development of lattice technology is not mature enough. On the contrary, the elliptic curve scheme is based on a mature theory and provides better security. However, most of the current ring signature schemes are optimized for internal protocols, and the research objective only focuses on a single transaction. In the case of large-scale transaction scenarios, there are obvious deficiencies in signature efficiency and storage space. We propose a representative ring signature scheme based on smart contracts. The major contributions of this work are presented below.

Multiparty secure computation is introduced to the ring signature to ensure that the signatory’s data participate in the calculation without being leaked.Smart contracts as trust endorsements are used to achieve the trusted interaction of transactions.The exterior of a ring signature is optimized to expand its research scope.

## 3. Preliminaries

### 3.1. Ring Signature

The structure of the ring signature comprises multiple keys. A few parameters of this structure form a ring association during the process of constructing the signature. The structure of the ring signature is presented in Figure 2. In this figure, pk denotes a public key set; s denotes a random array; c represents a generated array, named challenges; and the subscripts of different parameters represent the identity of the participants. The challenges generated by the participants form a ring association based on some predefined rules. When the participants are arranged in order, the computation process is mathematically expressed as follows:(1)$$\begin{array}{c} \;\;\;L_{i}=s_{i}\times G+c_{i}\times pk_{i}\\ R_{i}=s_{i}\times H_{0}\left(pk_{i}\right)+\\ c_{i+1}=H_{1}\left(m,L_{i},R_{i}\right) \end{array}c_{i}\times I$$
where G denotes the base point of the elliptic curve, m represents the signed data, $I=sk_{s}H_{0}\left(pk_{s}\right)$ represents the key image, and $H_{0}$ and $H_{1}$ represent two different hash functions, where $H_{0}$: $E\left(F_{q}\right)\to E\left(F_{q}\right)$, $H_{1}$: ${\left\{0,1\right\}}^{*}\to F_{q}{}^{*}$. Specifically, the challenge $c_{i+1}$ corresponding to the next participant i+1 is composed of the previous participant’s challenge $c_{i}$, the public key $pk_{i}$, and the random $s_{i}$, loop operation according to the above rules. When the last participant generates the challenge by adding its public key and random number, a challenge for the first participant is generated. Finally, a ring-like structure is formed. In this process, in order to be able to form a closed loop, it is necessary to start the signature from a private key, which is different from the public key. The challenge of private key generation is expressed as follows:(2)$$\begin{array}{c} L_{s}=u\times G\\ R_{s}=u\times H_{0}\left(pk_{s}\right)\\ c_{s+1}=H_{1}\left(m,L_{s},R_{s}\right) \end{array}$$
where $pk_{s}$ is the public key corresponding to the signature initiator. The private key owner generates the challenge by replacing the random numbers $s_{s}$ with random number u. In this process, the key owner does not need to use its own corresponding challenge $c_{s}$. The remaining participants generate the challenges based on Equation (1). When the private key owner also obtains the corresponding challenge, a closed loop is completed. Contrary to the other participants, the random number $s_{s}$ of the private key owner is not randomly generated. Instead, it is obtained by using the private key $sk_{s}$ and the random number u. This process is mathematically expressed as follows:(3)$$s_{s}=u-c_{s}\times sk_{s}$$

Now, the ring signature is completed, and the obtained signature is $\partial =\left(m,I,c_{0},s_{0},...,s_{n-1}\right)$.

### 3.2. Elliptic Curve

Suppose that there is a large prime number q and the integer field $F_{q}$ takes q as the module. Then, an elliptic curve is a set of points defined by $E_{q}\left(a,b\right)$ and satisfies $\Delta =\left(4a^{3}+27b^{2}\right)\mathrm{mod}q\neq 0$. Among them, $a,b,x,y\in F_{q}$, and $E_{q}\left(a,b\right)$ denotes a nonsingular elliptic curve. Specifically, $E_{q}\left(a,b\right)$ satisfies the following mathematical expression:(4)$$y^{2}=\left(x^{3}+ax+b\right)\mathrm{mod}q$$

If a point P(x,y) satisfies the $E_{q}\left(a,b\right)$ equation, then the point Q(x,−y) is a negative point of P, i.e., P=−Q. Let the points $P\left(x_{1},y_{1}\right)$ and $Q\left(x_{2},y_{2}\right)$ be the points on the elliptic curves $E_{q}\left(a,b\right)$ and P≠Q, then the line l passes through the point (P, Q) and intersects the elliptic curve at the point $R^{’}\left(x_{3},-y_{3}\right)$. The points of $R^{’}$ symmetrical about the x-axis are R and R=P+Q.

The points on the elliptic curve $E_{q}\left(a,b\right)$ and the infinite point O together form an additive cyclic group of prime order q as follows:(5)$$G_{q}=\left\{\left(x,y\right):a,b,x,y\in F_{q}\right\}$$

Now, the double-point operation defined on *G_q_* is:(6)$$kP=P+\ldots +P\left(k\in Z_{q}^{*}\right)$$

## 4. Materials and Methods

### 4.1. Design of Multiparty Secure Computing

Multiparty secure computing is a technique that enables multiple entities to simultaneously perform computations without compromising privacy. It uses the principle of multiparty secure computing to collect the opinions of all the participants in the ring signature. As presented in Figure 3, in multiparty secure computing, an organization needs to perform comprehensive computing using data. In this work, it uses smart contracts as the computing intermediaries, which are divided into two steps. First, the smart contract collects the public keys of all the participants. Then, it generates random numbers corresponding to the number of participants ($x_{i}$ denotes the random number of the corresponding participant, $\forall x_{i}\in N^{*}$) and encrypts them one by one by using the public key of the participant. The ciphertexts of the random numbers are sent to the respective participants. The participants use their own private keys to decrypt the random number ciphertext and process it according to the opinions of the participants. If they agree, the random number is increased by one. If they object or abstain, the random number remains unchanged. Finally, the processed random number is sent to smart contracts. After collecting the random numbers processed by all the participants, the smart contract is calculated to obtain the opinions of the participants. The computation method is mathematically expressed as follows:(7)$$v=N^{’}-N$$
where v denotes the number of entities who agree, $N^{’}$ denotes the sum of random numbers processed by all the participants, and N represents the sum of the generated random numbers. During the complete process, the smart contract does not directly interact with the participants and the data is stored in the blockchain. It is necessary to encrypt the random numbers generated by the smart contract before transmission. Please note that the random numbers processed by the participants can be transmitted in plain text. In this process, the participants do not know the random number processing of other participants. In this work, as a trusted third party, the smart contract does not disclose the user’s privacy data, and the computational results only show the overall opinion of the participants. Please note that the results do not present the detailed opinion of any participant. Therefore, the multiparty security computing method used in this work effectively hides the opinions of the participants and completes the collection of overall opinions.

### 4.2. Design of Representative Ring Signature

We use the improved ring signature, i.e., the representative ring signature, as the signature algorithm. The signature process in the improved ring signature algorithm is presented in Figure 4. The smart contract is the owner of the private key in the ring signature. In addition to smart contracts, the participants also include public key owners in the ring signatures. It should be noted that in the proposed method, the smart contract is also called the chaincode, and the private key in the ring signature is stored in the chaincode. The blockchain acts as a carrier of data interaction between smart contracts and the participants. The smart contract first initiates a ring signature event to obtain the public keys of all the members participating in the ring signature. Then, it uses multiparty secure computing to obtain the opinions of all the participants. Afterwards, a public–private key pair is generated in the smart contract. The private and public keys of all the participants are used to perform ring signatures. Finally, the signature is published in the blockchain to complete the signature.

During the complete ring signature process, the smart contract adopts one-time secret signature scheme. The public key of the smart contract is randomly generated, and other participants cannot obtain the public key. Each round of ring signature requires the public keys of all the participants. Therefore, it is impossible to forge the signature of a smart contract. The opinions of the participants collected by the smart contract are processed as follows:(8)$$\eta =\frac{v}{n}$$
where η denotes the approval proportion; v denotes the number of approvals; and n represents the number of participants, except the smart contract. The smart contract does not characterize the overall opinion. Instead, it adds the approval proportion η as a parameter to the signature to form a ring signature $\partial =\left(m,I,\eta ,c_{0},s_{0},...,s_{n-1}\right)$. In the signature verification stage, the approval proportion and preset coefficients are compared, and then the final opinion is determined, which is more in line with the actual requirements. Finally, as compared with the ring signature before improvement, the representative ring signature used in this work only requires one ring signature to complete the collection of participants’ opinions.

In order to understand the design scheme of this paper, we explain the whole process of using representative ring signatures by enumerating an example. We consider Alice’s election as chairman in a company as an example to illustrate the scheme. According to the rules, if the candidate obtains the affirmative vote of half or more of the board members, the candidate will become the chairman of the board. This round of voting is proposed to use a voting system based on the scheme designed in this paper. At the beginning, each board member registers an account on the platform and the system generates a unique public–private key pair for them. The supervisor initiates a voting event on the platform, and the board members upload their respective public keys on the system. Afterwards, internally (i.e., in a smart contract), a random number is assigned to each board member and encrypted using their uploaded public keys. Then, the encrypted random number is sent to each member. After the board members obtain the encrypted random numbers, they can obtain the original random numbers by decrypting them with their respective private keys. At this point, the “approval“ and “disapproval“ selection bar will pop up on the interface. If a board member chooses to approve, its corresponding random number will be automatically increased by one; otherwise, the random number corresponding to the board member will remain unchanged. When the selection is completed, the processed random numbers are uploaded in the system, i.e., smart contract. Based on the collected and assigned random numbers, the system calculates the number of votes in favor and combines them with the number of voters to obtain the proportion in favor. At this point, the system triggers the signature event. First, a set of public–private key pairs are randomly generated in the system. Then, all the public keys of the voters that have been obtained and the system-generated public–private key pairs are used as the parameters to invoke the preset ring signature algorithm for generating the signature and adding the approval proportion in the signature as the final signature result. Once the signatures are completed, the system sends them to each board member, who can enter the preset proportion, i.e., the minimum approval proportion to satisfy a successful campaign, such as ½, for signature verification. During the verification process, the system takes the public keys of all the signatures, including the public keys of smart contracts, and signatures as parameters, and invokes the signature verification algorithm to verify the validity of the signatures. It compares the size of the approval proportion and the preset proportion. If the approval proportion is greater than or equal to the preset proportion, it outputs the opinion as approval; otherwise, it outputs disapproval. Finally, the verification results and opinions are fed back to each board member. The supervisor announces whether Alice is the chairman of the board based on this voting opinion. This example explains that the representative ring signature has application value.

### 4.3. Representative Ring Signature FSM Model

We illustrate the representative ring signature process by using finite-state machine (FSM), which mainly includes two stages, namely signature and verification. Let the whole ring signature alliance chain system be represented by finite-state automata $M=\left(S,\sum ,f,S0_{},Z\right)$, where $S=\{S_{0},\ldots ,S_{15}\}$ denotes a set of signature node states; $S_{0}$ represents the initiating ring signature state, which is the only initial state; $S_{1}$ represents generated random number; $S_{2}$ represents encrypted random number; $S_{3}$ represents uploaded random number ciphertext; $S_{4}$ represents obtained random number ciphertext; $S_{5}$ represents decrypted random number ciphertext; $S_{6}$ represents processed random number; $S_{7}$ represents random number after upload processing; $S_{8}$ represents random number after acquisition processing; $S_{9}$ represents random number calculation; $S_{10}$ represents the ring signature; $S_{11}$ represents upload signature; $S_{12}$ represents the obtained signature; $S_{13}$ represents the verified signature; $S_{14}$ represents the display success; and $S_{15}$ represents the display failure. As presented in Figure 5, ∑={a,b} denotes an input alphabet; a represents “TRUE”; b represents “FALSE”; f represents the transition function, which is a function of S*∑ to S; and $Z=\left\{S_{14},S_{15}\right\}$ is the final state set.

The specific ring signature process is divided into four stages. The first stage is the ring signature initiation stage, in which the smart contract initiates a ring signature event. After successful initiation, a random number is generated and encrypted with the public key of the participant. The encrypted random number ciphertext is sent to the participant corresponding to the public key. The steps of the aforementioned process are repeated until all the participants have received the random number ciphertext. The second stage is the random number processing stage. In this stage, each participant first receives the corresponding random number ciphertext and decrypts it. The decrypted random number is processed by the participant in accordance with the situation. Finally, the participant uploads the processed random number to the smart contract. The third stage is the ring signature stage. The smart contract collects the random numbers processed by the participants, calculates the approval proportion, and performs the representative ring signature. Finally, the smart contract uploads the signature. The fourth stage is the signature verification stage. The verifier verifies the signature after obtaining it. If the verification succeeds, the signature is declared valid, and if the verification fails, the signature is declared invalid.

### 4.4. Algorithm Description

The ring signature algorithm consists of two parts, i.e., signature algorithm and verification algorithm. The signature algorithm is presented in Algorithm 1. The precondition is that the public keys of all the participants have been obtained in the smart contract and the number of voting opinions has been obtained through the multiparty security calculation. First, the number of positive comments is divided by the number of participants. The quotient is used to represent the percentage of approval. Then, the smart contract uses a key generation algorithm to obtain a public and private key pair, adds the public key to a public key group consisting of participants, and uses the private key as an argument to call a mirror function to generate a key mirror. Afterwards, the smart contract initiates the formal ring signature, starts processing with the private key, generates a random number u, and calculates the challenge c[s+1] of the next signatory by using Equation (2). The public keys, except for the smart contract, are processed one by one. When signing a public key, a random number s[i] is generated and calculated by using Equation (1) to obtain the next challenge. When the subscript of the challenge value reaches the length of the array, it is replaced with subscript 0 until the challenge c[s] corresponding to the subscript of the private key is calculated. At this point, all the public keys are signed. Based on the obtained challenge s[s], the random number c[s] corresponding to the private key is calculated using Equation (3). Finally, the content m of the signature, the result of the signature η, the generated c[0], and the random array s generated by all the participants are added to the signature, and the ring signature is completed.
**Algorithm 1** Ring Signature**Input:** pk[n−1],t**Output:** ∂1η=v/n2pks, sks ←KeyGen();3pk[n]←append(pk[n−1],pks);4I←GenKeyImage(pks);5⊳the initiator starts signing6u←rand.Int();7Ls=u*G8Rs=u*H0(pks);9c[s+1]=H1(m,Ls,Rs);10⊳the participants start signing11for i=1;i<n;i++ do12       index=(s+i)mod n;13       s[i]=rand.Int()*;*14       s[index]=s[i]*;*15       Li=si*G+ci*pki;16       Ri=si*H0(pki)+ci*I*;*17       c[i+1]=H1(m,Li,Ri)*;*18       if i==n−1 then19           c[s]=c[i]20**          else**21           c[(index+1) mod n]=c[i];22**          end if**23**   end for**24   s[s]=u−c[s]*sks*;*25   ∂=(m,I,η,c[0],s[0],⋯,s[n−1])*;*26   Return ∂;

The verification algorithm is simpler as compared to the signature algorithm. It is mainly used to verify if there is at least one private key in the ring signature to ensure its validity. A function of signature opinion determination has been added in this work. The process is shown in Algorithm 2. The input parameters include the ring signature ∂ and the preset proportion p. First, the random array s is obtained from the ring signature and challenge is computed using Equation (1). It starts from c[1], repeatedly calculates c[n] in order, and assigns it to c[0], where n denotes the length of array s. It compares the preset proportion p with the approval proportion η in the signature to obtain the signature opinion. Finally, the consistency of c[0] is verified using the signature. The verification result and signature opinion are used as the output, and the process of verification ends.
**Algorithm 2** Signature Verification**Input:** ∂,p**Output:** msg,result=(TRUE,FALSE)1ring=∂.s2for i=1;i<ring.size;i++ do3       Li=si*G+ci*pki*;*4       Ri=si*H0(pki)+ci*I*;*5      c[i]=H1(m,Li,Ri)*;*6      if i==ring.size−1 then7           c[0]=c[i]*;*8      else9           c[i+1]=c[i];10      end if11end for12 if ∂.η<p then13        msg=“against”;14 else15msg=“approve”;16end if17result=Equal(∂.c[0],c[0];18Return msg, result;

## 5. Results and Discussion

### 5.1. Experimental Environment

In this work, we use the Hyperledger Fabric platform to perform testing and use Docker containers to deploy a four-node network in a cloud server. The solo mode in the test environment is adopted for the consensus algorithm. The environment configuration includes a 64-bit Ubuntu 16.04 LTS operating system with 2-core CPU, 2 GB memory, 1 Mbps broadband, Golang programming language, and Fabric v1.2.0.

### 5.2. Experimental Design

In order to test the performance of the scheme proposed in this paper, we compare the performance of the representative ring signature scheme with the ECC ring signature scheme [33] in terms of the size of the signature, the speed of the signature, and the speed of verification. First, based on the design idea of the ECC ring signature scheme, we use Golang language to reproduce the code. At the same time, the representative ring signature scheme is also encoded in the Go language. Both of them choose the same elliptic curve and hash function, i.e., secp256k1 elliptic curve and sha3-256 hash function. Five sets of experiments are designed as follows:When the number of participants changes, test the signature size of the two methods;When the number of participants changes, test the signature time of the two methods;When the number of participants changes, test the verification time of the two methods;When the number of participants is 5, the signature time of different times is tested;When the number of participants is 5, the verification time of different times is tested.

Based on the experimental design, it is observed that the parameters to be adjusted are the private key used for ring signature, the public key group of the participants, and the number of rounds for ring signature. The data in the first three groups of experiments are randomly generated by the key generation algorithm. The data in the last two groups of experiments are generated by setting circular statements. For convenience, the experiment is designed using the web service interface and tested by using the stress-testing tool Apache ab. In the aforementioned experiment, the test data of the second and third groups fluctuate. In the aforementioned experimental scheme, the test data of the second and third groups fluctuate. In order to reduce the error, it uses the average of 10 test data samples as the final result.

### 5.3. Evaluation Methods

We divide the test data into two dimensions, namely spatial dimension and time dimension. In case of the former, when there is only one participant, a signature of size a is generated. Afterwards, the size of the signature increases b times, when a participant is added, i.e., a public key is added. Now, the ECC ring signature contains n signature size for a member, presented as:(9)$$A_{1}=a+\left(n-1\right)b$$

In a round of opinion collection, each member generates a signature once. Therefore, the total signature size in a round of signatures is mathematically expressed as:(10)$$A_{2}=nA_{1}=na+n^{2}b-nb$$

The improved ring signature collects the opinions of all the participants and submits them to the smart contract for signature. Therefore, only one ring signature is needed in a round of signatures. However, the smart contract also participates in the signature. As compared to a signature of same specifications, the number of participants representing the ring signature is one more than the total participants. The signature size of n members is expressed as:(11)$$A_{3}=a+nb$$

In a ring signature, the number of public keys cannot be less than three, so $n\geq 3,n\in N^{*}$. $A_{2}>A_{3}$ can be obtained by using (10) and (11). As n increases, the gap between the two ring signature schemes increases in terms of storage space.

In this work, the signature time and verification time are metrics for measuring time. The ring signatures used in the two schemes are based on the elliptic curve problem. The time complexity of the signature algorithm based on the elliptic curve problem is based on the modular multiplication operation. The time complexity of a modular multiplication operation is mathematically expressed as:(12)$$W=O\left(w^{2}\log w\right)$$
where w denotes the data size of modular multiplication operation. In this work, within a certain range, when the number of participants changes, w does not change significantly. Therefore, when different participants sign, the time of modular multiplication is regarded as unchanged. Let the duration of a single public key signature be $t_{1}$, and the duration of a single verification be $t_{2}$. Now, when there are n public key signatures, the ECC ring signature needs to be signed and verified n times. Therefore, the duration of a single signature is $T_{1}=nt_{1}$, and the duration of a single verification is $T_{2}=nt_{2}$. However, for a representative ring signature, for any public key to participate in the signature, it only needs one signature and one verification. Therefore, its single signature duration is $T_{3}=t_{1}$, and single verification duration is $T_{4}=t_{2}$.

The performance comparison between the scheme proposed in this paper and the ECC ring signature scheme is shown in Table 1.

### 5.4. Experiment and Analysis

Based on the comparative analysis of multiple dimensions, a more comprehensive understanding of the advantages and disadvantages of the two methods mentioned above is obtained. Table 2 and Figure 6 present a comparison chart of the signature size, when the signature participants change. The expression presented in Equation (10) shows that the ECC ring signature increases with the increase in the number of the participants. In addition, the growth rate of the signature size also increases. The expression presented in Equation (11) shows that as the number of participants increases, the size of the ring signature increases evenly. It can also be observed from the figure that as the number of signing participants increases, the gap between the signature sizes of the two schemes becomes larger and larger. This shows that the increase in the ECC ring signature is considerably high, while the increase in the representative ring signature is stable. In this experiment, the increase is fixed at 128. Therefore, the representative ring signature performs better than the ECC ring signature in terms of signature size.

The speed of signature in ring signatures is an important indicator of the performance of an algorithm. In some scenarios, where the ring signatures are applied, such as voting and election, the speed of signature affects the overall efficiency of the system. Table 3 and Figure 7 present a comparison chart of the speed of a single signature, when the number of participating ring signatories changes. In a round of signatures, the number of ECC ring signatures is related to the number of participants, which means that each additional participant needs to sign one more time. On the contrary, the representative ring signature has nothing to do with the number of the participants, which means that no matter how many participants are added, only one signature is required. The expression presented in Equation (12) shows that within a certain range, the speed of a single signature is independent of the number of participants. Therefore, the signature time representing the ring signature in the figure jumps up and down around a value. This error comes from network fluctuations. The signature time observed in the ECC ring signature is proportional to the number of participants. Therefore, the signature speed of the proposed representative ring signature is significantly faster as compared to the ECC ring signature. This advantage further increases with an increase in the number of participants.

The verification duration of a ring signature is also an integral part of evaluating the overall performance of an algorithm. As shown in Table 4 and Figure 8, the duration of a single verification of a ring signature is less than that of a signature. As the number of participants increases, the verification time of the ECC ring signature method also increases. Please note that this increase is almost the same each time a participant is added. In contrast, the verification duration of the proposed representative ring signature method varies around 85 ms. It is noteworthy that the verification duration is not affected by the changes in the number of participants. Considering Figure 7 and Figure 8, it is evident that the signature and verification time of a ring signature is not directly affected by the change in the number of participants. Instead, it is affected by the change in the number of signatures caused by the change of participants. The ECC ring signature technique shows that the signature time and verification time are proportional to the number of participants. The representative ring signature is a single signature regardless of the number of participants. Therefore, the experimental data of the signature duration and verification duration vary minutely. The analysis of the signature duration and verification duration of the representative ring signature verifies the expression presented in Equation (12), i.e., within a certain range, the data scale of the modular multiplication operation is hardly affected by the change in the number of participants.

The aforementioned analysis shows that the representative ring signature scheme performs better as compared to the ECC ring signature scheme in terms of signature and verification efficiency. In order to further illustrate the performance gap between the two schemes in terms of signature and verification, and to supplement the experimental results from another perspective, two groups of experiments are designed to compare the signature and verification durations of the two schemes by changing the number of signature rounds when the participants are fixed to five. Figure 9 shows the comparison chart of signature duration when the number of signature rounds changes. Figure 7 shows that when the number of participants is five, the single signature duration of the ECC ring signature scheme is approximately five times that of the representative ring signature. Moreover, Figure 9 shows that with the increase in the number of signature rounds, the curves of the two schemes show a linear increase, and the slopes of the two schemes are roughly 5 times different, i.e., for the same number of signature rounds, the corresponding longitudinal coordinate values differ by a factor of 5.

By fitting the data of the signature duration comparison experiment, it is evident that when the number of participants is five, the signature efficiency of the ring signature scheme is five times higher as compared to the ECC ring signature scheme. In this work, when the number of participants is five, the validation times of the two schemes are fitted and analyzed. As shown in Figure 10, the changes in the trend of the curve are consistent with those shown in Figure 8. Both curves are proportional to the number of signature rounds, and the ratio of their slope values is close to 5. Please note that the verification time of the two schemes is approximately equal when the participants tested in Figure 8 are equal to five. The duration of a single verification is multiplied by the number of signature rounds. Considering Figure 9 and Figure 10, the signature durations of the two schemes are still influenced by the number of signatures. As the number of rounds increases, the number of signatures increases exponentially. Based on the multidimensional data analysis, it is evident that the representative ring signature is superior to the ECC ring signature scheme in terms of signature size, signature efficiency, and verification efficiency.

### 5.5. Safety Analysis

A secure ring signature scheme should satisfy three aspects, including correctness, unconditional anonymity, and unforgeability. As compared to the traditional ring signature algorithm, in this work the signatures are initiated by smart contracts and the opinions of the participants cannot be disclosed during the signing process. Therefore, the unconditional anonymity analysis in this study needs to be replaced with a confidentiality analysis.

#### 5.5.1. Correctness Analysis

Assuming that the signature $\partial =\left(m,I,c_{0},s_{0},...,s_{n-1}\right)$ Initialization: Given a security is generated based on Algorithm 1 and the verifier verifies it based on the ring signature verification algorithm, then for n increasing from 0 to n−1, the verifier performs the following calculations. $L_{i}=s_{i}\times G+c_{i}\times pk_{i}$, $R_{i}=s_{i}\times H_{0}\left(pk_{i}\right)+c_{i}\times I$, $c_{i+1}=H_{1}\left(m,L_{i},R_{i}\right)$. When i≠s, then $c_{i+1}=H_{1}\left(m,L_{i},R_{i}\right)$. When i=s, from Equations (2) and (3), we can deduce Equations (13)–(15).
(13)$$\begin{array}{l} L^{’}{}_{s}=s_{s}\times G+c_{s}\times pk_{s}\\ =\left(u-c_{s}\times sk_{s}\right)\times G+c_{s}\times pk_{s}\\ =uG-c_{s}\left(sk_{s}\times G\right)+c_{s}\times pk_{s}\\ =uG-c_{s}\times pk_{s}+c_{s}\times pk_{s}\\ =uG=L_{s} \end{array}$$
(14)$$\begin{array}{l} R^{’}{}_{s}=s_{s}\times H_{0}\left(pk_{s}\right)+c_{s}\times I\\ =\left(u-c_{s}\times sk_{s}\right)\times H_{0}\left(pk_{s}\right)+c_{s}\times I\\ =u\times H_{0}\left(pk_{s}\right)-c_{s}\times sk_{s}\times H_{0}\left(pk_{s}\right)+c_{s}\times I\\ =u\times H_{0}\left(pk_{s}\right)-c_{s}\times I+c_{s}\times I\\ =R_{s} \end{array}$$
(15)$$c^{’}{}_{s+1}=H_{1}\left(m,L^{’}{}_{s},R^{’}{}_{s}\right)=H_{1}\left(m,L_{s},R_{s}\right)$$

The above derivation shows that the signature satisfies $c_{0}=c_{n}$. Therefore, the scheme in this paper satisfies the correctness.

#### 5.5.2. Unforgeability Analysis

**Theorem** **1.***If the elliptic curve discrete logarithm problem (ECDLP) is difficult, then**the ring signature scheme designed in this paper is unforgeable*.

**Proof.** Assume that in the random oracle model, the attacker ℜ adaptively chooses the messages to attack. The challenger Ꞇ receives a random instance (P,aP) of the discrete logarithm problem, with the objective of computing the value of a. The challenger Ꞇ sets the public key of the signatory $U_{*}$ to $pk_{i^{*}}=aP$. Ꞇ acts as a subroutine of ℜ and plays the role of ℜ’s challenger. In order to ensure generality, we assume that all the queries are different. Now, we show how the challenger Ꞇ responds to the attacker ℜ’s query.

(1)Initialization: Given a security parameter k, the challenger Ꞇ runs the initialization algorithm to obtain the system parameters. Then, the challenger Ꞇ sends the system parameters to the attacker ℜ.(2)Hash query: The challenger Ꞇ has an L
list (ui,xi), which is initially empty. When the attacker ℜ makes a public key query, the challenger Ꞇ chooses a random value xi∈ℤq*, and sets pki=xiP. Then, the challenger Ꞇ adds (ui,xi) to the list and returns pki to ℜ.(3)User public key query: The challenger Ꞇ has an Lu
list (ui,xi), which is initially empty. When the attacker ℜ performs a public key interrogation on ui, the challenger Ꞇ chooses a random value xi∈ℤq*, sets pki=xiP, and then adds (ui,xi) to the list and returns pki to the attacker ℜ.(4)Private key query: When the attacker ℜ
queries the user for the private key, if ui=ui*, then Ꞇ stops the operation. Otherwise, the challenger Ꞇ returns the corresponding private key xi to ℜ.(5)Ring signature query: The attacker ℜ
submits the message m and the public key set R of n users. The challenger Ꞇ outputs the ring signature ∂. If the user’s public key pks∈R satisfies pks≠pki*, then the challenger Ꞇ executes the signature algorithm to reply with a signature ∂, where the user corresponding to the public key pks is the real signatory. Otherwise, the challenger Ꞇ performs the following steps:
1.Randomly select si, u∈ℤq*, and calculate:
(16)Li={si×G+ci×pkii≠su×Gi=s
(17)Ri={si×H0(pki)+ci×Ii≠su×H0(pki*)i=s
(18)ci=H1(m,Li,Ri)2.Finally, the ring signature for message m is output as ∂*=(m,Is*,c0,s0,...,ss*,...,sn−1).
(6)Forgery: The attacker ℜ outputs the signature of another message m* for the signatory ui*. By forking lemma [35], choosing a different hash function H1, the challenger Ꞇ can obtain the following two valid signatures with the same value ss:
(a)∂=(m,Is,c0,s0,...,ss,...,sn−1)(b)∂*=(m,Is*,c0,s0,...,ss*,...,sn−1)


Hence, the following two equations are true.
(19)$$s_{s}=u-ac_{s}$$
(20)$$s_{s}=u^{’}-ac^{’}{}_{s}$$

From Equations (19) and (20), we can obtain $a=\left(u-u^{’}\right)/\left(c_{s}-c^{’}{}_{s}\right)$. The attacker ℜ can successfully solve $a=sk_{s}$ for the given instance (P,aP).

Assuming that an attacker ℜ successfully forges a valid signature with a non-negligible probability, there is an algorithm Ꞇ that can successfully solve the ECDLP problem with a non-negligible probability. However, ECDLP is considered to be a difficult problem. Therefore, the probability of the ring signature constructed in this paper is negligible under the random oracle model, that is, the representative ring signature satisfies the unforgeability. □

#### 5.5.3. Confidentiality Analysis

In this work, we use multiparty security computing to collect the signature opinions from the participants. During the collection process, the opinions of the participants are hidden in the random number. The third party can only know the opinions of the participants by obtaining the random number plaintext before and after the signature opinion processing. As the random number before the signature opinion processing is encrypted, the opinions of the participants are not known to anyone other than themselves. Therefore, the scheme proposed in this work satisfies confidentiality.

## 6. Conclusions

The ring signature provides indistinguishable signature authentication. It plays an important role in identity protection and makes up for the lack of user identity protection in the blockchain. The improved ring signature scheme proposed in this work generates smaller signatures in a round of signatures and provides faster signature and verification speeds. The proposed improved ring signatures take advantage of the collaborative work based on multiparty secure computations and the fairness and transparency of smart contracts. In this work, multiparty security computation is used to collect the opinions of the participants, and the smart contract represents the ring signature for one-time filling. This effectively ensures the security of the signature, making the improved ring signature more suitable for large-scale ring signature applications. Moreover, we aim for the external optimization of ring signature, which can further improve the performance of the ring signature by transplanting the best current ring signature algorithm. However, the application range of the ring signature scheme proposed in this work is smaller as compared to the traditional ring signature. In addition, it is only applicable to the scenarios where the identity of the nodes is known and the nodes cannot enter and leave the network freely, such as consortium blockchain or private blockchain. At present, the scheme proposed in this paper does not support the public blockchain. In this regard, the proposed scheme still needs further improvements.

## Figures and Tables

**Figure 1 sensors-22-06805-f001:**
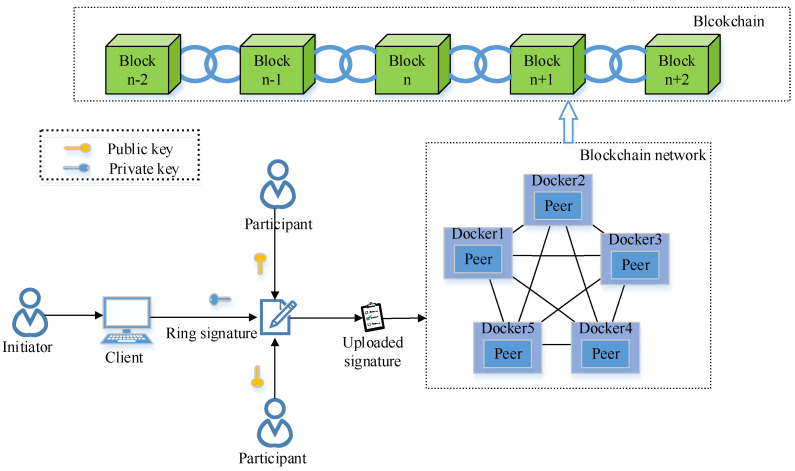
Ring signature basic operation flow.

**Figure 2 sensors-22-06805-f002:**
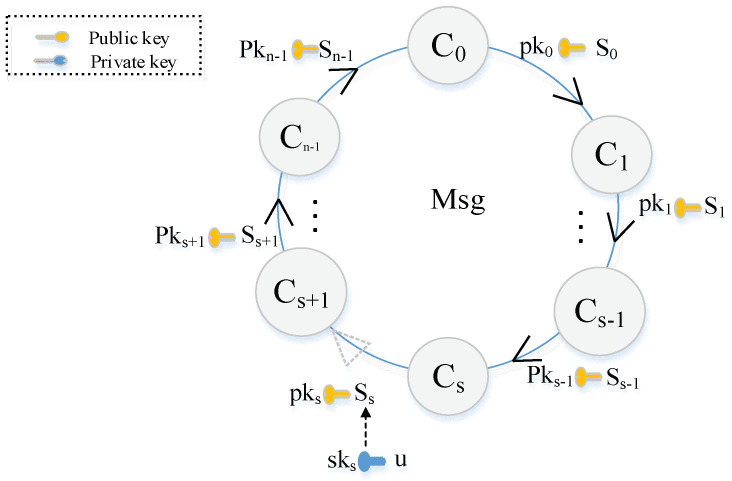
The structure of ring signature.

**Figure 3 sensors-22-06805-f003:**
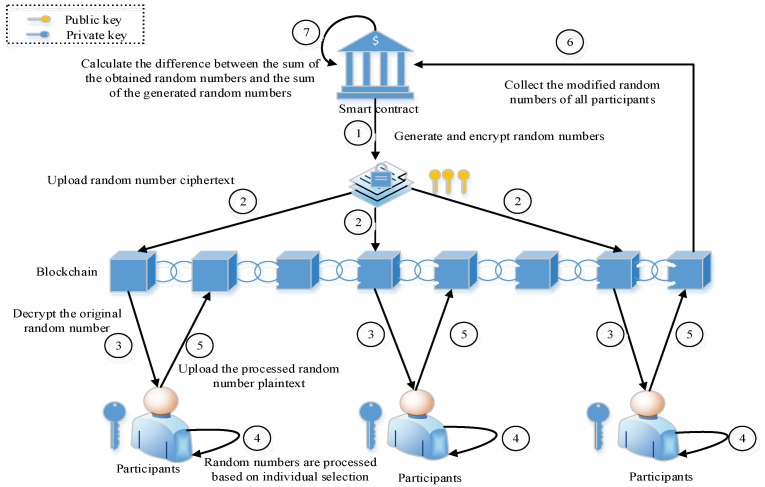
A flowchart of multiparty security computation technique.

**Figure 4 sensors-22-06805-f004:**
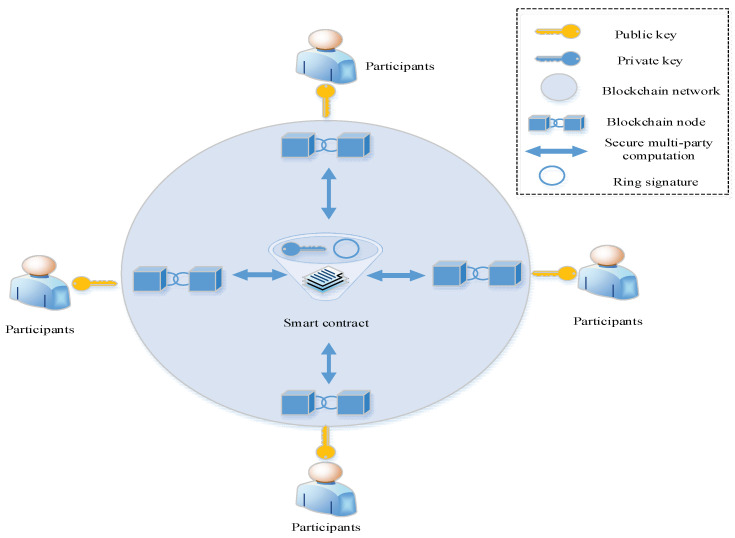
A representative ring signature.

**Figure 5 sensors-22-06805-f005:**
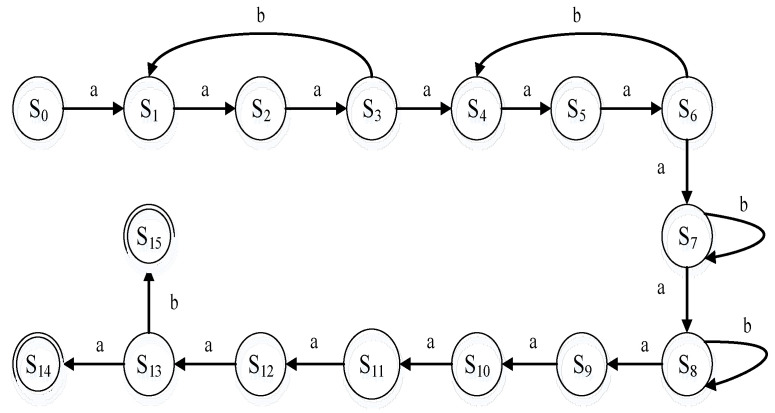
The FSM model of a ring signature.

**Figure 6 sensors-22-06805-f006:**
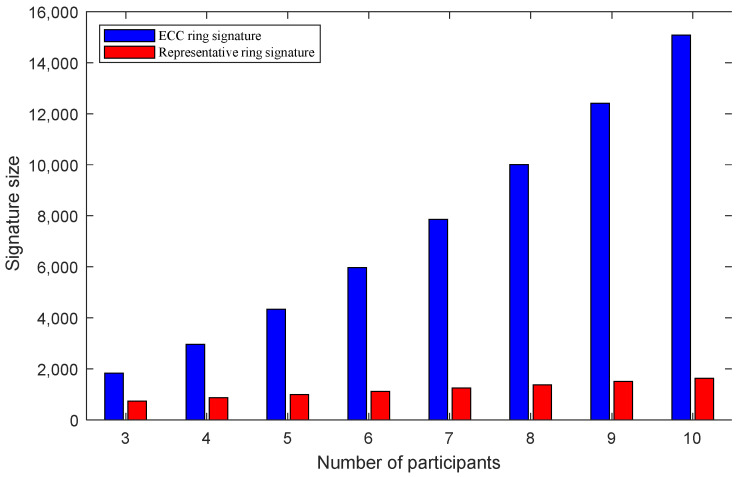
The comparison of signature sizes.

**Figure 7 sensors-22-06805-f007:**
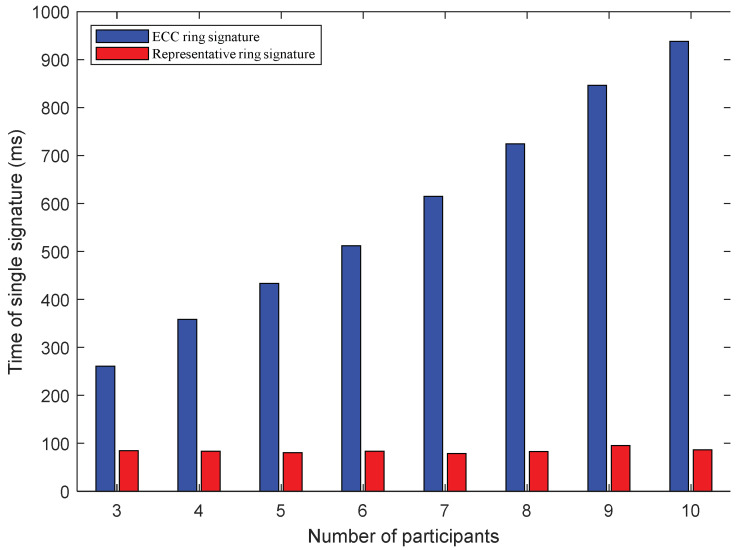
The comparison of single signatures in terms of computational speed.

**Figure 8 sensors-22-06805-f008:**
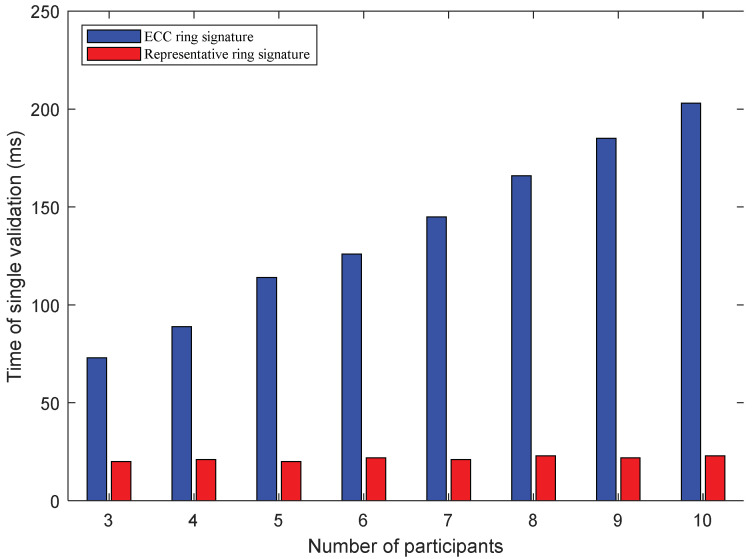
The computational speed comparison for a single validation between two schemes.

**Figure 9 sensors-22-06805-f009:**
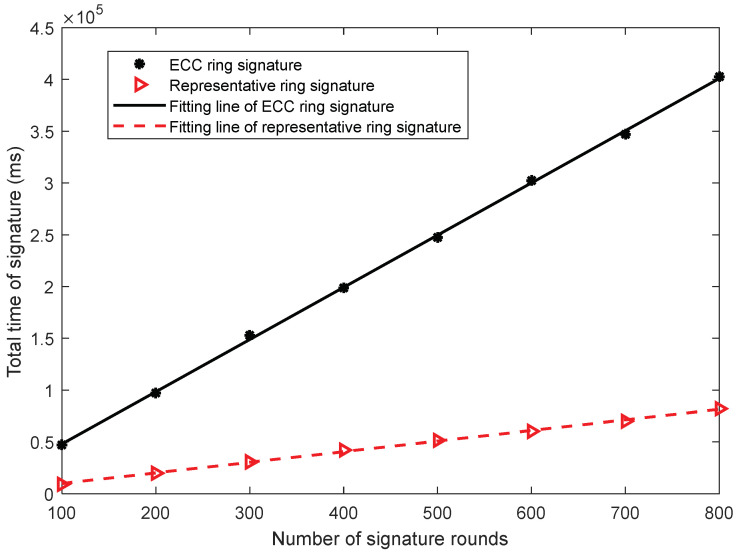
The total time comparison of signatures between two schemes.

**Figure 10 sensors-22-06805-f010:**
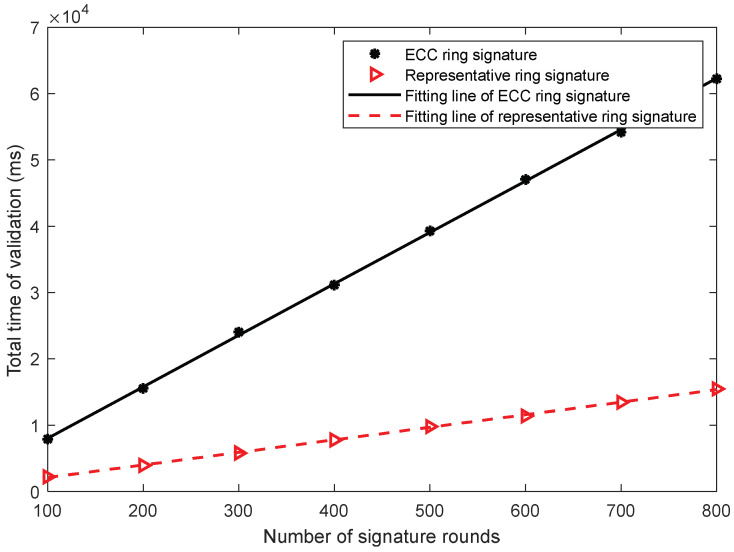
The total time comparison consumed for signature validation.

**Table 1 sensors-22-06805-t001:** The performance comparison of two schemes.

Scheme	Signature Time	Verification Time	Signature Size
ECC ring signature	$nt_{1}$	$nt_{2}$	$na+n^{2}b-nb$
Representative ring signature	$t_{1}$	$t_{2}$	a+nb

**Table 2 sensors-22-06805-t002:** A comparison of signature sizes of two schemes.

Scheme	Number of participants							
	3	4	5	6	7	8	9	10
ECC ring signature	1836	2960	4340	5976	7868	10,016	12,420	15,090
Representative ring signature	740	868	996	1124	1252	1380	1508	1637

**Table 3 sensors-22-06805-t003:** Comparison of signature time of two schemes (unit: ms).

Scheme	Number of Participants							
	3	4	5	6	7	8	9	10
ECC ring signature	261	359	434	512	615	724	846	938
Representative ring signature	85	84	81	84	79	83	96	87

**Table 4 sensors-22-06805-t004:** A comparison of signature verification time of two schemes (unit: ms).

Scheme	Number of Participants							
	3	4	5	6	7	8	9	10
ECC ring signature	73	89	114	126	145	166	185	203
Representative ring signature	20	21	20	22	21	23	22	23

## Data Availability

The data presented in this study are available on request from the corresponding authors.

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
