# Peer review of "Representative Ring Signature Algorithm Based on Smart Contract"

_sensors, 2022, doi:10.3390/s22186805_

Round 1

Reviewer 1 Report

I carefully observed the article "Research and practice of representative ring signature algorithm based on smart contract" and found suitable for publication after changes as suggested below.

1. Figure 1 should not contain any significant information so it shoul dbe updated or removed.

2. authors should add the significance of stud.

3. the comparison of study with literature is missing. authors needs to add bench mark table by comparing their results with other and should be add in discussion section.

4. Title of manuscript look not appropriate. it should be change some suitable title.

Reviewer 2 Report

1. The article does not disclose the question whether a public blockchain or a private one can be used for this scheme.

2. I would like to clarify the examples of the application of the king signature algorithm based on smart contracts

3. I would like to see the connection diagram of the participants (dockers and nodes) of the experiment.

Reviewer 3 Report

This paper proposes a ring signature algorithm based on smart contracts, which uses smart contract to organize the signature and prevent signature forgery via "one encryption per signature" strategy. The topic selected in this paper is new. However, there are still some issues as follows: 1. Some symbols are confusing, for example, 1) Hp should be a hash function to generate a big integer point of the elliptic curve. 2) The variable si-1 in equation (1) is lower-case, whereas the one in the description is upper-case. 3) What does "Ps" in the equation (2) mean? 4) The paper has the statement “finite state automata (S, ,f , S0,Z )”, What does "S0" mean? 5) What does the function “append” in Algorithm 1 mean? 6) What does the variable p in Algorithm 2 mean? ……
2. The paper said “The smart contract has a private key.” Where does this private key store? 3. I suggest adding objects to Algorithm 1-6. For example in the line 9 of Algorithm 1, if “who checks Expiry” is added may be more clear. 4. The proposed scheme has no attacker model and security model. 5. The proposed scheme lacks formal proof.

Round 2

Reviewer 3 Report

Although the authors have responded to the comments, there are still two suggestions as follows:

1. The authors should examine the full text again, there are still symbol errors. For example, Hp is still used in equation (2).

2. The author should prove the security of the proposed scheme under the random oracle model.
